# Reproductive Senescence in the Pollinator, *Megachile rotundata*

**DOI:** 10.3390/insects16060612

**Published:** 2025-06-10

**Authors:** Jacob B. Pithan, Brooke L. Kohler, Arun Rajamohan, Kendra J. Greenlee

**Affiliations:** 1Department of Biological Sciences, North Dakota State University, Fargo, ND 58102, USA; brooke.l.kohler@und.edu; 2USDA-ARS, Fargo, ND 58102, USA; arun.rajamohan@usda.gov

**Keywords:** aging, pollinator, solitary bee, foraging behavior, oocyte development, parental investment, reproduction

## Abstract

Pollinators play a crucial role in agriculture, yet we know little about how aging affects their performance. In this study, we examined whether aging reduces the foraging and reproductive ability of female alfalfa leafcutting bees, a solitary species widely used for pollinating crops. Based on patterns seen in other animals, we expected that older bees would show signs of aging, such as reduced foraging and lower-quality offspring. However, we found no evidence of such declines within our observed timeframe. Older females foraged more frequently and efficiently, resulting in larger food provisions and bigger offspring. We also observed no decline in reproductive organs with age, suggesting these bees maintain strong reproductive capacity throughout this timeframe. Our findings provide evidence that aging may not always result in reduced performance and suggest that older alfalfa leafcutting bees may be especially valuable for crop pollination. This work improves our understanding of aging in insects and highlights the importance of considering bee age when managing pollinator populations. Promoting the presence of older bees during peak flowering periods could enhance pollination and support agricultural productivity.

## 1. Introduction

The disposable soma theory postulates that organisms have a finite amount of resources to allocate between reproduction, maintenance, growth, storage, and other activities [1]. As organisms age, acquiring resources may become harder, as shown by age-related declines in foraging [2,3]. Additionally, as organisms age, they may incur additional maintenance costs, such as the repair of oxidative damage [4]. These additional costs and reduced ability to acquire specific resources may cause resources to be reallocated from other biological functions, resulting in a physiological trade-off (reviewed in [5]). These trade-offs are automated and unconscious, resulting in the rebalancing of existing physiological processes. This can lead to declines in reproductive function and fertility as an organism ages, which is known as reproductive senescence (reviewed in [6]).

Across taxa, patterns of reproductive senescence vary drastically (Table 1). Even within Insecta, reproductive plasticity is highly variable both between and within species (reviewed in [7,8,9,10]). In *Drosophila melanogaster*, as flies aged, carbonylated protein, a measure of oxidative damage, increased in both soma and eggs, and their offspring had decreased survival [11]. Similarly, in honey bee (*Apis mellifera*) embryos from older queens were smaller and had higher mortality rates compared to those of younger queens [12]. However, contrary to these studies, maternal age does not influence offspring survival in bean weevils (*Acanthoscelides obtectus*; [13]) or spined spider bugs (*Podisus maculiventris*; [14]). Additionally, different species exhibit varying foraging behavior with age. For instance, the paper wasp, *Ropalidia marginata*, decreases its foraging duration with age, but shows an increase in foraging efficiency, as older wasps return with more food in less time. This improvement may be attributed to enhanced spatial familiarity and learning [15]. In contrast, *A. mellifera* foragers older than 29 days experience a general decline in flight performance [16]. To fully understand the consequences of aging and its influence on reproductive fitness, it is essential to explore the full diversity of life-history strategies, sociality, and abiotic factors found in insects that leads to this variability [7].

One such insect species, the solitary alfalfa leafcutting bee, *Megachile rotundata*, provides an opportunity to examine consequences of aging on its reproduction. This species is a commercial pollinator that is extensively managed in agriculture practices [92]. Females experience an energetically demanding adult stage as the sole caretaker of their nests, in contrast to social bees with a worker caste. Like others in the Megachilidae family, females build nests consisting of linear arrays of leaf-covered brood cells within both natural and artificial cavities. After selecting a suitable cavity, a female lines its interior with leaf pieces to create a cell. She then fills the cell with pollen and nectar, lays an egg, and finishes by adding more leaf pieces to seal it (see [93]; Figure 1). This process is repeated until the cavity’s entrance is reached, which she caps (capping) with additional layers of smaller leaves (reviewed in [92,94]). They construct 8–12 brood cells in their lifetime, each requiring 14–15 leaf discs that they cut out using their mandibles (reviewed in [94]). Provisioning is performed only by females, requiring greater energetic investment compared to social bees that share the responsibility of care for the young. Furthermore, females spend the majority of the day flying an average of 5–6 h [95]. At the end of the season, the offspring undergo facultative diapause and overwinter [85,96,97].

Studies on reproduction and senescence in *M. rotundata* are limited. Certain behaviors, such as male harassment [98] and longer foraging trips [99], have been linked to lower fecundity. Flight performance has also been shown to decline with age in females and is speculated to impact fecundity [100]. In one study, older females showed a slight reduction in ovariole size, though they were not observed under natural field conditions [51]. Additionally, Royauté et al. [93] found both individual and age-dependent variation in nest construction and offspring provisioning. However, as they only sampled a single nest per female, they concluded that a more thorough investigation would require comparing cell provisioning across multiple nests, including both early and late ones.

Reproductive success in solitary bees is thought to be limited by nest location, food supply for offspring, and the number of oocytes produced by females [101]. Given the substantial parental investment made by the female *M. rotundata* for their offspring, it is anticipated that there will be a decline in reproductive investment as they age. However, further research is needed on this concept of reproductive capacity and aging in a solitary bee using a longitudinal study under true field conditions.

Field studies on reproductive senescence in natural insect populations are rare, but they are crucial for gaining a comprehensive understanding of how reproduction changes with age (reviewed in [102]). Additionally, these field studies are complicated due to individual variation that leads to differences in behavior, life-histories, and reproductive fitness [103,104]. In our field study, we established a correlation between the age of *M. rotundata* and reproductive senescence by monitoring individual changes in foraging behavior over time. We achieved this using marked females and nesting boxes equipped with GoPro cameras. We also directly tested the effects of aging on reproductive senescence, including assessing changes in oocyte size, as well as various aspects of foraging behavior, such as the frequency, type, and duration of foraging trips, provision size, and reproductive output (number of nests and brood cells), and offspring quality (survivorship, dry mass, and diapause status). Given the additional energy expenditure that solitary bees experience [98,105,106,107,108,109], we hypothesized that as *M. rotundata* females age, both foraging activity and reproductive fitness decline.

## 2. Materials and Methods

### 2.1. Rearing and Marking Bees

Loose brood cells containing *M. rotundata* prepupae were purchased from JWM Leafcutter, Inc. (Nampa, ID, USA) in the spring of 2023. Prepupae were separated into 24-well plates and kept at 6 °C with 30% relative humidity in constant darkness for approximately 3 months until used in the experiments. During the months of June and July, prepupae were removed from storage and placed into an incubator (Precision Scientific, New York, NY, USA) kept at 29 °C with 21% relative humidity in constant darkness to initiate metamorphosis. Upon emergence as adults, female *M. rotundata* were cold-anaesthetized and marked on the thorax with two colored dots using oil-based paint (Sharpie, Newell Brands, Atlanta, GA, USA) for identification.

### 2.2. Ovarian Development

To assess ovarian development, recently emerged (<24 h) marked females were released with males at nest boxes in Fargo, ND (46.94°, −96.86°). Eight nest boxes were placed 100 m apart along a drainage ditch that was adjacent to an alfalfa field. At each nest box, 50 marked females and 25 males were released. In 7-day intervals (7, 14, 21 days), female *M. rotundata* were collected from the nest boxes (*n* = 20 from the 8 nest boxes at each interval), brought to the lab, and preserved in 70% ethanol until dissected. Additionally, a subsample (*n* = 24) of recently emerged females were also immediately preserved for dissection. Prior to dissection, thorax width was measured using digital calipers (General Tools, Secaucus, NJ, USA). Using Cohan–Vannas spring scissors (Fine Science Tools Inc., Foster City, CA, USA), the sterna from the abdomen were removed. Once the abdominal cavity was open, the ovaries were then extracted intact in phosphate-buffered saline and placed between a slide and coverslip. The length and width of the oocytes of each ovary were measured on a Lionheart LX Automated Microscope (Agilent Technologies, Santa Clara, CA, USA) using a GFP filter (excitation 469 and emission 525 nm). For each female (*n* = 84), the volumes of the two basal and the 10 largest primordial oocytes were estimated using the equation *V* = (*πr*^2^) (*L* − 2*r*) + (4/3) (*πr*^3^) for basal oocytes and *V* = (4/3*πr*^2^) (*L*/2) for primordial oocytes, where *r* is the radius and *L* is the length of the oocyte [110].

### 2.3. Foraging Behavior

To assess age-dependent changes in foraging behavior, the number of trips, the duration of trips (min), type of foraging (leaf, provisioning, or capping), and number of brood cells constructed were measured throughout an individual female’s lifetime. Using the same field site previously described, eight nest boxes were outfitted with GoPro cameras (GoPro Hero7, GoPro™, San Mateo, CA, USA) and BlinkX time-lapse controllers (CamDo Solutions, Vancouver, BC, Canada; Figure 1A). We released 50 marked females and 25 males at each box. Of the eight nest boxes, seven were successfully established. On days 1, 7, 14, and 21, the GoPro cameras recorded the nesting behavior at each successfully established box for 14 h (06:00 to 20:00) at a frame rate of 120 fps and a resolution of 1080 p. For each female, the number, length, and type of foraging of each trip were recorded that day. Only females that were observed on all 4 days (*n* = 18) were used in the data analysis. From the videos, we were able to determine the different phases of nesting (cell construction, provisioning, and cell completion) and calculate the number of brood cells constructed that day.

### 2.4. Offspring Quantity and Quality

During the foraging experiment, nests were collected weekly from the boxes and brought back to the lab. The nests (*n* = 174) were x-rayed upon arrival and again 2 weeks later (UltraFocus 23X29 Digital Specimen Radiography System, Hologic Inc., Marlborough, MA, USA). From the two x-rays, the number of offspring and their provision size and diapause status were determined. Provision size was calculated from the images using ImageJ (version 1.53o, National Institutes of Health, Bethesda, WI, USA). Provisions were cylindrical, so we used the equation = πr^2^h to calculate the volume. Only the brood cells that had no apparent consumption (*n* = 901) were used in data analysis. After the second x-ray, individual brood cells were placed into 24-well plates based on their diapause status. Offspring that were undergoing direct development were placed into an environmental chamber set at 29 °C (Precision Scientific, Bohemia, NY, USA). Offspring that had entered diapause were placed into a 6 °C environmental chamber (Conviron, Winnipeg, MB, Canada) for the overwintering period (August 2023–March 2024) and then transferred to 29 °C in April 2024 to continue their development. Once emerged, sex was determined and adult offspring (*n* = 1028) were dried to a constant weight at 50 °C for 2 days (UF30 Universal Oven, Memmert, Eagle, WI, USA).

### 2.5. Statistical Analyses

Statistical analyses were performed in JMP (version 17.0.0, SAS Institute, Cary, NC, USA). After determining a normal distribution and equal variance, mixed model ANOVAs with Tukey HSD post hoc tests were used to compare the means for provision size and offspring dry weight. Due to non-normal distribution, oocyte volume (basal and primordial), number of foraging trips, duration of foraging trips, number of brood cells constructed, number of nests, and number of brood cells were analyzed using a Kruskal–Wallis test with Wilcoxon method for nonparametric comparisons between pairs. For number and duration of foraging trips, each foraging type (leaf, provision, or capping) was analyzed independently. Offspring survival, diapause status, and the sex of the offspring were analyzed using binomial logistic regressions. *p*-values less than 0.05 were considered statistically significant. Data are represented as means ± 95% confidence intervals throughout.

## 3. Results

### 3.1. Ovarian Development

The basal and primordial oocyte volumes were significantly affected by age (Figure 2; Basal: X^2^_3,163_ = 100.698, *p* < 0.0001; Primordial: X^2^_3,819_ = 205.86, *p* < 0.0001). Upon emergence, females had the smallest basal and primordial oocytes. By day 7, there was an 8-fold increase in basal oocyte volume (0.211 ± 0.033 mm^3^) and a 3-fold increase in primordial oocyte volume (0.021 ± 0.003 mm^3^). By day 14, basal, but not the primordial, oocyte volume continued to increase (0.276 ± 0.040 mm^3^). Following day 14, both oocyte volumes were maintained.

### 3.2. Foraging Behavior

We analyzed age-related changes in foraging activity, including the number, duration, and type of foraging trip, for 18 females (Figure 3). Females made 190 ± 22 trips in total over the 4 days. During the first day in the field, females made no foraging trips and spent the day exploring the nest box and the surrounding area. On the 7th day, a majority of females began foraging around 08:00, but 33% did not start foraging until after 10:00. Additionally, 7-day-old females stopped foraging earlier (~17:00) than the 14- or 21-day-old bees that were still foraging when the cameras stopped recording at 20:00. Observations of the type of foraging throughout the day appeared to show no synchronization among individuals. No capping was observed on day 7.

There was a significant effect of parent age on the number of brood cells under construction (Figure 3D; X^2^_3,17_ = 48.512, *p* < 0.0001). During the first day out in the field, no females constructed brood cells. By day 7, 50% of females worked on 2 brood cells. As the females aged (day 14), there was a significant increase in the number of brood cells under construction, with 72% of females working on 3 brood cells in 1 day. There was no significant difference in the number of brood cells being worked on when comparing day 14 to day 21.

As females aged, more foraging trips for capping, leaf, and provisions were made (Figure 4A; X^2^_3,17_ = 15.916, *P_Capping_* = 0.0012; X^2^_3,17_ = 49.073, *P_Leaf_* < 0.0001; X^2^_3,17_ = 49.893, *P_Provision_* < 0.0001). Seven-day-old females on average made 0 capping, 15.4 ± 6.7 leaf, and 20.4 ± 4.4 provisioning trips. By day 14, the number of leaf and provisioning trips being made increased by 75% and 79%, respectively. Additionally, day 14 was the first day nest capping was recorded, with 33% of the females on average making 12.44 ± 10.19 capping trips to complete their nest. There were no significant changes in the number of foraging trips for capping, leaf, and provisions when comparing 14- to 21-day-old females.

As females aged, the duration of leaf and provision foraging trips shortened (Figure 4B; X^2^_3,17_ = 137.48, *P_Leaf_* < 0.0001; X^2^_3,17_ = 142.873, *P_Provision_* < 0.0001), but the duration of capping trips remained consistent (X^2^_2,17_ = 0.316, *p* = 0.574). Capping trips took females on average 108.20 ± 12.70 s. Seven-day-old females took the longest time to complete leaf (324.86 ± 51.79 s) and provision trips (826.28 ± 61.16 s). By day 14, there were 58% and 26% decreases in the length of time needed to complete leaf and provision trips, respectively. By day 21, there was no change in the duration of leaf trips, but a continued decrease in the time needed to complete provision trips (36% decrease when compared to 7-day-old females).

### 3.3. Offspring Quantity and Quality

There was a significant effect of parent age on the number of nests (Figure 5A; X^2^_3,7_ = 10.140, *p* = 0.017) and brood cells (Figure 5B; X^2^_3,8–87_ = 12.260, *p* = 0.007). During the first week, females completed the fewest number of nests and brood cells. During the second week, the number of nests and brood cells peaked with over a 4-fold increase per observed individual. There was no significant difference in the number of nests or brood cells between the second, third, and fourth week.

There was a significant effect of parent age on the number of brood cells within each nest (Figure 6; X^2^_3,173_ = 10.489, *p* = 0.015), but not the number of empty brood cells (X^2^_3,173_ = 0.758, *p* = 0.860). Of the seven nests collected during the first week, we found that on average there were 8.125 ± 0.536 brood cells within the nests, with 0.625 ± 0.766 brood cells being empty. Of the 87 nests collected during the second week, we found that there was no significant difference in the number of brood cells within the nests, and only 4.6% of nests contained empty brood cells. By the third week, there was a significant decrease in the average number of brood cells within a nest (6.95 ± 0.413) when compared to the first week. There was no significant difference in the number of brood cells within a nest during the third and fourth week.

Offspring diapause status, survival, and sex were affected by parent’s age (Figure 7A–C; X^2^_3,1105_ = 32.894, *P_Diapause_* < 0.0001; X^2^_3_,_1098_ = 11.24, *P_Survival_* = 0.021; X^2^_3,1028_ = 9.351, *P_Sex_* = 0.025). Overall, there was a continuous increase in diapause-destined offspring, with 82% of the offspring being destined the first week and 98% during the fourth week. Although offspring survival varied with parent age (ranging from 90 to 98%), the fluctuations were not consistently directional, that is, they did not show a clear pattern of increase or decrease as parents aged. During the first week, 44% of the offspring were females. There was no difference in sex ratio from the first week to the third week. By the fourth week, there were significantly more females (9%) when compared with the first week; however, this ratio was not different from the second and third week.

Provision volume and the dry weight of offspring were affected by parent age and the offspring sex, but not their interaction (Figure 7D,E; F_3,900_ = 9.583, *P_Provison_
_(age)_* < 0.001; F_1,900_ = 7.932, *P_Provision_
_(sex)_* = 0.005; F_3,900_ = 2.527, *P_Provision_
_(age*sex)_* = 0.056; F_3,1026_ = 23.353, *P_Weight_
_(age)_* < 0.0001; F_1,1026_ = 16.583, *P_Weight_
_(sex)_* < 0.0001; F_3,1026_ = 1.168, *P_Weight_
_(age*sex_*_)_ = 0.321). Regardless of parent age, female offspring were consistently given a bigger provision and weighed more compared to males. During the first week, female offspring were given a provision size of 110.35 ± 4.87 mm^3^ and weighed 11.45 ± 0.88 mg. There was no change in provision size or dry weight of females between the first and second week; however, by the third week, there was a significant increase (11%) in female weight. There was no change in provision size or weight of females during the fourth week. During the first week, males were given a provision size of 96.13 ± 5.06 mm^3^ and weighed 9.59 ± 0.52 mg. There was no change in male provision size from the first to third week, but by week four, there was a 6% increase (compared to the first week). Male weight did not change from the first to second week, but increased by 2% (compared to the first week) during the third. There was no difference in male weight between the third and fourth week.

## 4. Discussion

Reproduction imposes a toll on organisms, requiring significant energy expenditure. However, both between and within species, the cost varies due to resource availability, resource acquisition, life-history traits, and sociality (reviewed in [5,38]). Unlike social bees, in solitary bee species, such as *M. rotundata*, individual females are responsible for collecting pollen and nectar for themselves and offspring provisions, constructing nests, provisioning for offspring, and guarding their offspring. Given the additional energy expenditure, we hypothesized that *M. rotundata* would experience physiological trade-offs leading to declines in reproductive fitness as they age. However, contrary to our hypothesis, *M. rotundata* did not experience apparent reproductive senescence within our experimental timeframe. Additionally, we found that as females aged, they increased their foraging rates, which led to larger provisions and thus bigger offspring.

### 4.1. Ovarian Development

Similarly to a study conducted by Richards [51], we found that female *M. rotundata* have a pair of ovaries of comparable volume, each consisting of three ovarioles with oocytes. The oocytes experienced age-related increases from when females first emerged to 21 days. However, unlike Richards [51], we did not see a decrease in basal oocyte size later in life, which is likely due to differences in experimental design. In Richards’ [51] experiment, females were reared in cages with no mating, nest construction, or restricted foraging, whereas in our experiment, females were allowed to age under more natural conditions and thus did not experience extended lifespans and declines in oocyte volume. Parallel to Richards [51], we found that the main increase in oocyte volume (basal and primordial) occurs about 7 days after emergence and is likely due to access to pollen. Pollen is the main source of proteins and lipids for most solitary and social bees [106,111,112,113]. Several studies have found that pollen is essential for reproduction in bees [63,64,114,115], and without pollen, female insects can gradually resorb oocytes [116,117,118]. The second increase in basal oocyte volume that occurred after 7 days was likely due to the establishment of routine and an increase in the number of foraging trips. It has been documented previously [51,119] and was observed in this study that *M. rotundata* females spend the majority of the first week mating, exploring shelters, selecting suitable tunnels for nest construction, orientating themselves with the surroundings, and constructing and provisioning their first brood cell. By the second week, the number of foraging trips being made peaked. Females likely consumed additional nectar and pollen to support the continued development of reproductive tissues. Solitary bees rely on their foraging success to gather resources that are crucial for not only provisioning their offspring, but also their own nourishment. Additionally, access to abundant resources can lead to increases in egg size and number and the overall health of reproductive tissue [114,120]. We believe that the absence of a decline in reproductive tissues indicates that *M. rotundata* optimize their reproductive capacity during the first 21 days of life. Reproductive senescence has been well documented in social insects; however, as also shown in the current study, solitary insects often exhibit little to no reproductive senescence (reviewed in [8,9]). Social insects, such as ants, termites, and honey bees, have complex social structures that depend on long-lived reproductive individuals (e.g., queens). This social structure creates an evolutionary pressure for extended reproductive capacity that eventually leads to reproductive senescence and the replacement of the reproductive individual. Solitary insects like *M. rotundata* lack such a social system, and therefore, the selective pressure for extended reproduction lifespan is absent.

### 4.2. Foraging Behavior

Age-related changes in insect foraging behavior are well documented and can vary significantly across species (reviewed in [8,10,121]). In this study, not only did we find no evidence of age-related declines in foraging performance in *M. rotundata*, but females actually improved, at least during the time period that we observed them, taking more foraging trips at faster rates, leading to more brood cells. During the late summer (mid-July to mid-August), we found that *M. rotundata* began foraging around 08:00 (approximately 2 h after sunrise), with the majority of individuals stopping foraging a few minutes before 20:00. However, it is important to note that we did observe some females making foraging trips after 20:00. In the morning, the initial foraging trips were frequently lengthy, lasting an average of 34 min, with certain flights extending beyond 90 min. Similarly to this study, Klostermeyer [96] found that first flights were lengthy and *M. rotundata* spent much of that time basking in the sun. Surprisingly, we found that 7-day-old females had a shorter foraging window compared to 14- or 21-day-old females. We speculate that the shorter window of foraging in younger bees is due to physiological changes in flight muscles, with *M. rotundata* first needing to build up their flight muscles after emerging as adults, as shown in a variety of insects, including fruit flies [122], mosquitoes [123] dragonflies [124,125], hawk moths [126], and honey bees [16,127].

The observed maximum number of brood cells under construction in 1 day was four. However, during the 3rd and 4th weeks, most females completed two brood cells and began working on a third in 1 day. This result is on par with other studies looking at *M. rotundata* [66,96] and other solitary bees, such as *Osmia lignaria* [128,129] and *Osmia imaii* [130]. The majority of solitary bees produce a single provisioned brood cell each day (reviewed in [131]) and generally mature their next oocyte in a 22–24 h span [128]. From our study, we found that two oocytes (one basal oocyte per ovary) were enlarged. To lay a third egg within a day, *M. rotundata* must be able to quickly mature the next basal oocyte in less than 12 h (our window of observed foraging). This rapid enlargement of oocytes and the provisions needed for offspring would require extensive foraging.

In this study, we found that the number of foraging trips increased in the same pattern (increase from day 7 to day 14 and no difference at day 21) as the number of brood cells constructed. This result makes sense because as females make more foraging trips, they can provision and construct more brood cells. This age-related increase in the number of foraging trips is likely due to a decrease in the duration needed to complete a foraging trip. Similarly to this study, honey bees exhibit a significant increase in the number of foraging trips over the course of 8 days, coupled with shorter flight durations, likely due to learning [132]. Likewise, the increase in *M. rotundata’s* foraging rate following the first week most likely reflects the benefits of experience, learning, and memory. In Hymenoptera, including ants [2], honey bees [127,132,133], bumble bees [134,135], and carpenter bees [65], learning is the major contributor to age-related increases in foraging performance. It has been postulated that learning and experience in honey bees can improve their navigation, reward collection from flowers, and efficiency at moving between flowers and plants [132,136]. However, this would require further study in *M. rotundata*.

When examining the foraging behavior of female *M. rotundata*, there is variability in both the timing of specific foraging activities and the number of trips required to finish a brood cell. In this study, we found *M. rotundata* needed 11–25 provisioning trips to complete brood cells, with the average provisioning rate being 136–325 s per provisioning trip (age-dependent). Among solitary bees, the number of foraging trips needed to provision a brood cell ranges widely, from 2 to 40 trips (reviewed in [131]). Additionally, Gathmann and Tscharntke [137] found that across eight solitary bee species, the average provisioning rates took around 300–1600 s. While it appears that *M. rotundata* is capable of making provisioning trips at faster rates than most solitary species, it is important to note that factors including floral abundance, foraging distance, and weather conditions, as well as offspring size, sex, and, notably, age, as evidenced by this study, can impact foraging (reviewed in [138,139,140]).

### 4.3. Offspring Quantity and Quality

In insects, age-related changes are thought to significantly impact both the quantity and quality of offspring, as well as the offspring sex and amount of parental investment. Generally, as insects mature, the number of offspring they produce tends to increase initially, peaking at a certain age before declining [3,17,18,19]. Additionally, as they age, some species exhibit less parental investment like provisioning [55,56] and guarding against usurping females and predators [57]. This shift in reproductive strategy with age underscores the trade-offs insects face between quantity and quality of offspring. In other megachilid bees, the relationships between parental investment, offspring sex ratios, and size have been documented in *Anthidium septemspinosum* [58], *Osmia bruneri* [59], *O. cornifrons* [56], *O. cornuta* [55], *O. lignaria* [19], and *O. rufa* [3]. In contrast to the previous studies, we found no evidence of directional changes in sex ratios or decreases in provision size or offspring weight. The reduction of parental investment and emphasis on male production has been hypothesized to be largely due to seasonal transitions of available flower resources in the field [19,59]. Alternatively, other studies attribute these trade-offs to aging, such as the loss of scopa hairs, which diminishes the pollen-carrying capacity [56,58]. In this study, the higher provisioning rate is likely due to the age-related increase in foraging trips. However, it is also important to consider that the diapause incidence and provision size may be correlated. *M. rotundata* offspring that are diapause-destined have been shown to receive more provisions than non-diapausers [141,142]. Although we observed a correlation between parent age and diapause status, seasonality also plays a role in diapause occurrence in various species (reviewed in [143,144]), including *M. rotundata* [145,146]. Because we had a small number of non-diapausing offspring later in the experiment, we did not have enough statistical power to assess whether provision size increases for both non-diapausing and diapausing offspring. Furthermore, to determine if parent age affects offspring diapause status rather than seasonal factors, future experiments will need to include parents of different ages concurrently.

## 5. Conclusions

To the best of our knowledge, this is the first study to investigate individual changes in foraging with age in solitary bees. Contrary to studies of social bee species, we found no compelling evidence for reproductive senescence in the agriculturally important pollinator, *M. rotundata*. Additionally, we found that as females age, they become more efficient at foraging, provide bigger provisions, and have more and larger offspring that do not differ in survivorship. This age-related improvement in foraging efficiency is likely due to changes in flight muscle physiology, experience, and learning in *M. rotundata*, but further experiments are needed to elucidate the mechanisms behind the age-related changes we observed. Despite our conclusion of high parental investment, *M. rotundata* appears to maximize their reproductive capacity with the capability of laying more eggs than most solitary bees. This, coupled with their foraging performance, provides additional support for *M. rotundata* as a popular alternative pollinator. Overall, our results suggest that pollinator age should be taken into consideration when making pollinator management decisions, with the presence of older bees during peak bloom potentially providing greater rates of pollination.

## Figures and Tables

**Figure 1 insects-16-00612-f001:**
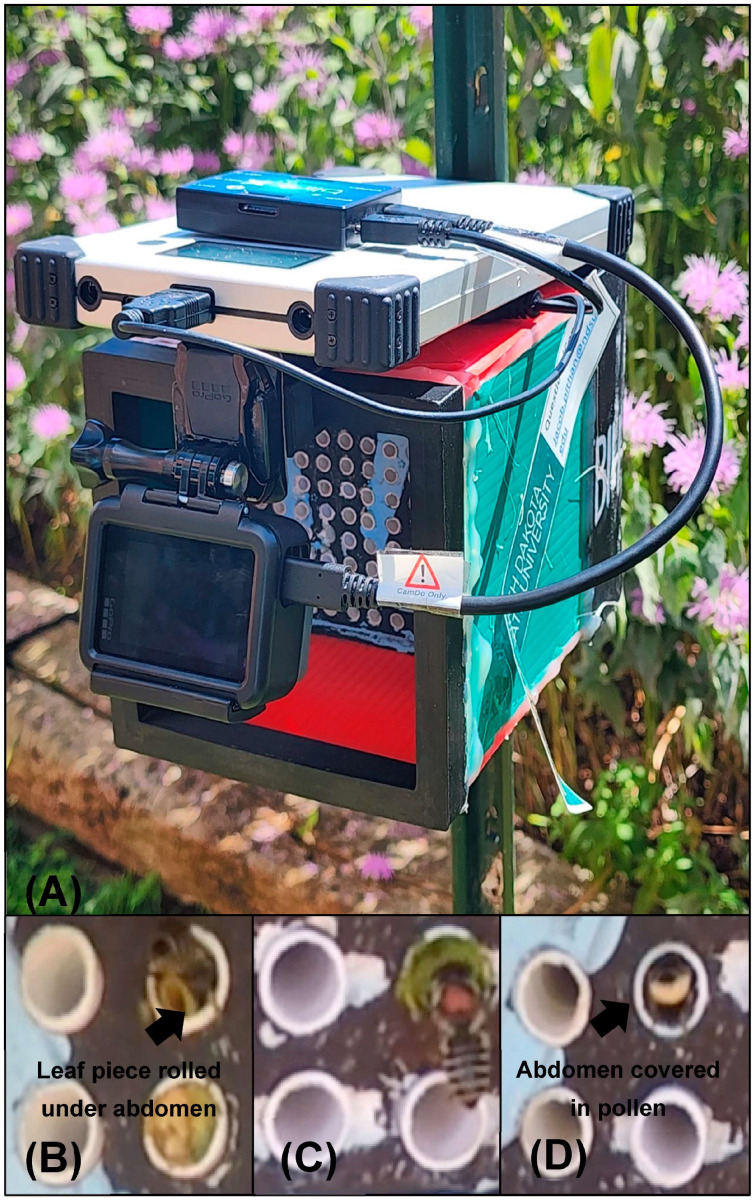
Nest box equipped with a GoPro camera, battery, and BlinkX time-lapse controller (**A**). Recording of foraging activities occurred from 6:00 AM to 8:00 PM. The type of foraging activity: leaf (**B**), cap (**C**), or provisioning (**D**) was identified based on the recordings.

**Figure 2 insects-16-00612-f002:**
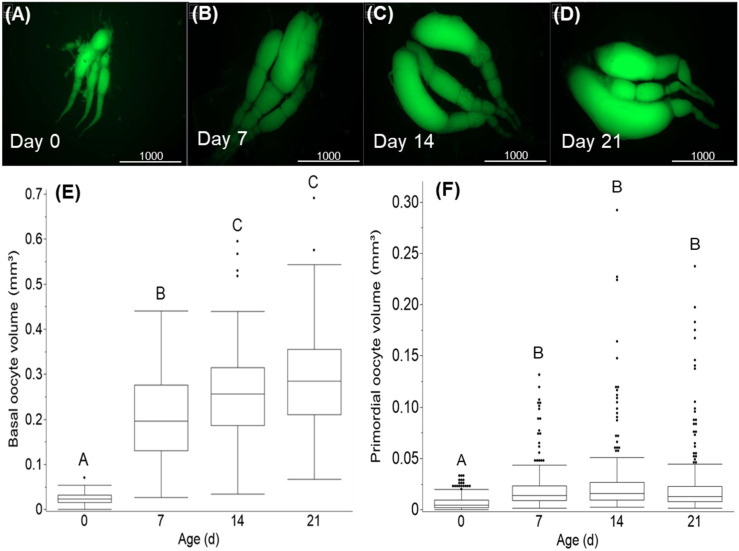
(**A**–**D**) Morphological changes in ovaries with age. Ovaries are composed of three ovarioles containing the basal and primordial oocyte. Scale bars correspond to 1000 µm. (**E**) Basal oocyte volume increased with age (P_Basal_ < 0.0001; *n* = 40–44 per age group). (**F**) Primordial oocyte volume increased with age (P_Primordial_ < 0.0001; *n* = 200–220 per age group). Ages with different letters were significantly different (*p* < 0.05).

**Figure 3 insects-16-00612-f003:**
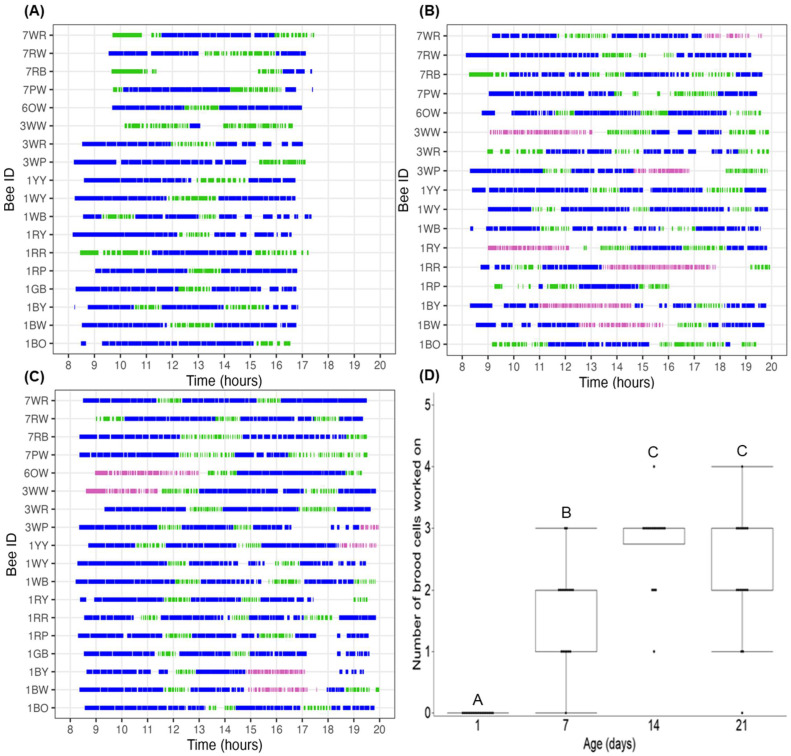
(**A**–**C**) Foraging activity of females (*n* = 18) on the observed days (days 7 (**A**), 14 (**B**), 21 (**C**)). Each row is an individual bee with the colored line segments representing a foraging trip and the length of each segment being correlated to duration of the trip. Colors represent the type of foraging: provisions (blue), leaf (green), capping (pink) from 6:00 am to 8:00 pm. No foraging occurred on day 1 (panel not shown), and no capping (pink) was observed on day 7. (**D**) The number of brood cells under construction was significantly affected by the age of the female (*p* < 0.0001). Ages with different letters were significantly different (*p* < 0.05).

**Figure 4 insects-16-00612-f004:**
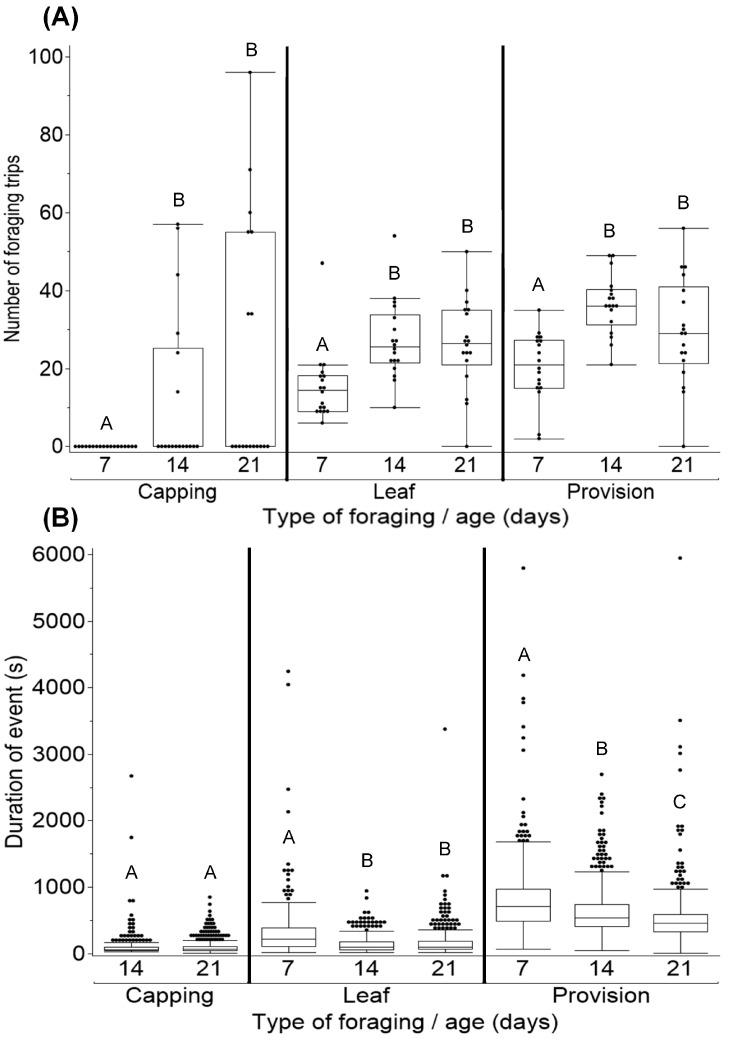
Number (**A**) and duration (**B**) of foraging trips made by females (*n* = 18). Foraging trips were separated by type, then analyzed. No capping was observed on day 7. There was a significant increase in number of capping trips (*p* = 0.0012) but no change in duration (*p* = 0.574) as females aged. Older females made more leaf (*p* < 0.0001) and provision (*p* < 0.0001) trips at faster rates compared to younger bees (*P_Leaf_* < 0.0001; *P_Provision_* < 0.0001). Ages with different letters were significantly different (*p* < 0.05).

**Figure 5 insects-16-00612-f005:**
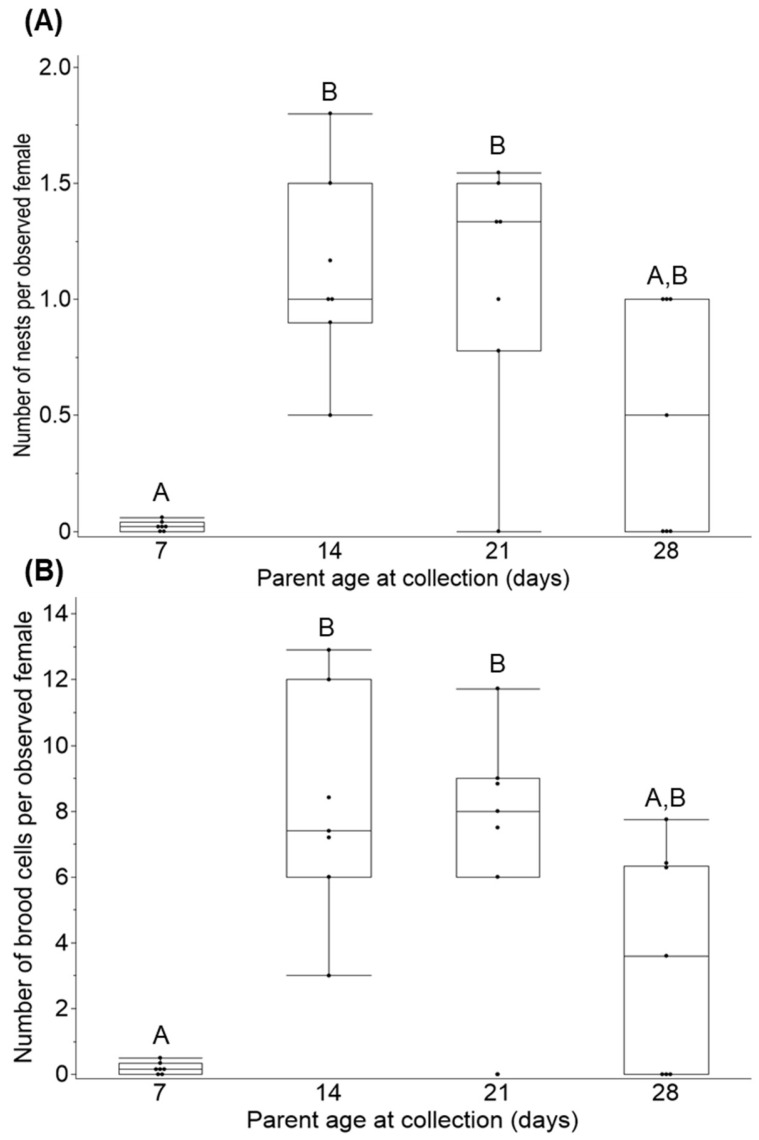
Number of nests (**A**) and brood cells (**B**) per observed female at nesting box (*n* = 8). There was a significant increase in the number of nests (*p* = 0.007) and brood cells (*p* = 0. 0065) as parents aged. Ages with different letters were significantly different (*p* < 0.05).

**Figure 6 insects-16-00612-f006:**
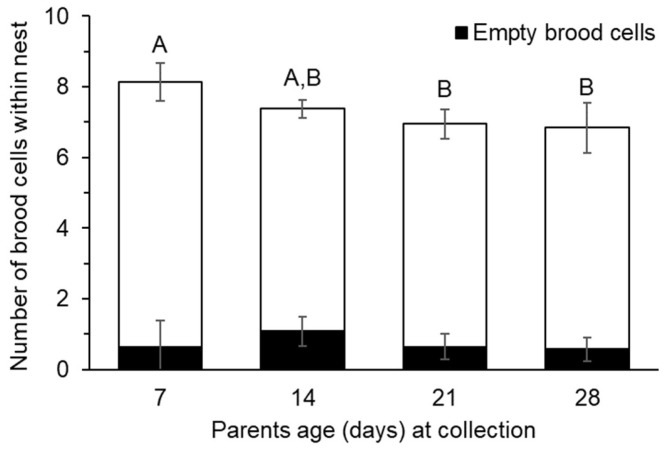
Average number of empty and completed brood cells within nests (*n*= 8–87 per age group). Parent age had a significant effect on the number of brood cells within nest (*p* = 0.015), but not the number of empty brood cells (*p* = 0.860). Ages with different letters were significantly different (*p* < 0.05).

**Figure 7 insects-16-00612-f007:**
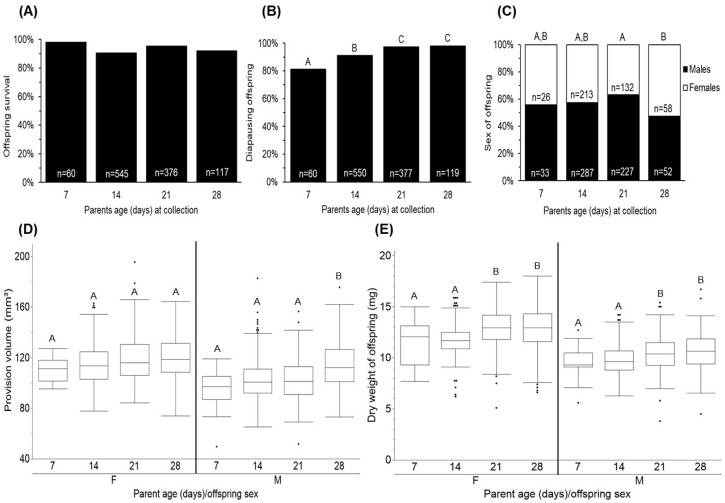
Age-related changes in offspring survival (**A**), diapause status (**B**), sex (**C**), provision volume (**D**), and dry weight (**E**). Parent age had no significant effect on offspring survival (*p* = 0.284). The occurrence of diapause in offspring increased with parent age (*p* < 0.0001). Parent age (*P_Provision_* < 0.0001; *P_Weight_* < 0.0001) and offspring sex (*P_Provision_* = 0.005; *P_Weight_* < 0.0001) had a significant effect on provision volume and offspring dry weight. Older parents gave a larger provision and had bigger offspring. Females were given a larger provision and weighed more than males. Ages with different letters were significantly different (*p* < 0.05).

**Table 1 insects-16-00612-t001:** Studies investigating age-related changes in male and female reproduction.

Female Reproductive Aspects	Decreased	Remain Unchanged	Increased
Fecundity	[3,17,18,19,20,21,22,23,24,25,26,27,28,29,30,31,32,33,34]	[35,36,37,38]	[39]
Offspring survival	[21,30,37,39,40,41,42,43,44,45,46,47]	[36,46,48]	
Breeding success	[49]	[49,50]	
Oocyte quality	[11,51,52,53]	[54]	
Parental investment	[23,48,55,56,57,58,59,60,61,62]		[63,64,65]
Mating willingness	[47]		[66,67]
**Male Reproductive Aspects**	**Decreased**	**Remain Unchanged**	**Increased**
Courting performance	[21,68,69,70,71]		[72]
Sperm motility	[73,74,75,76]	[77,78,79]	[80]
Sperm count	[74,79,81,82,83]		[77,84]
Sperm viability/quality	[77,85,86,87]	[79,83,88]	[84]
Pre-copulatory male competition	[72,78]		
Mating rates	[68,72,89,90]	[67]	[91]

## Data Availability

The raw data supporting the conclusions of this article will be made available by the authors upon request.

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
