# Peer review of "Reproductive Senescence in the Pollinator, Megachile rotundata"

_insects, 2025, doi:10.3390/insects16060612_

Round 1
Reviewer 1 Report
Comments and Suggestions for Authors
REVISION
No errors or erroneous considerations detected.
COMMENT
The manuscript examines the main parameters of an important wild bee, Megachile rotundata, also bred for its use as a pollinator. In general, the honey bee and some wild bees are considered to be less efficient as pollinators and as producing offspring with senescence. Experiments using special nest boxes on individuals purchased at prepupae stage in brood cells, placed near an alfalfa field, have made it possible to assess these and other parameters with many types of analysis.
The research is interesting because it examines in detail factors reporting in other papers negative effects of senescence on apoidea. The manuscript instead highlights different results for this species. A positive effect of senescence emerges, due to increased experience of individuals and physiological changes. These results are also important for the pollination service.
The text is very well written and without detectable errors; the references are relevant and numerous. Artificial intelligence may have been used in its processing?
Author Response
Review 1 did not have any comments that required changes to the manuscript.
Reviewer 2 Report
Comments and Suggestions for Authors
Dear Authors,
The findings on age-dependent reproduction and foraging activities of M. rotundata will contribute to the literature.
Some minor layout errors are noticeable:
Figure 1 should be centered
Figure 2 extends beyond the page, Please correct the Figure 3 title.
Please review the reference numbering system.
Best regards,

Author Response
Comment 1: Figure 1 should be centered
Response 1: This issue will be addressed by uploading high resolution figures.
Comment 2: Figure 2 extends beyond the page, Please correct the Figure 3 title.
Response 2: As above and the Figure title has been amended.
Comment 3: Please review the reference numbering system.
Response 3: Changes have been made to the manuscript.
Reviewer 3 Report
Comments and Suggestions for Authors
Aging of solitary bees is interesting case beside with honey bee for the topic of aging in general. Authors described several new original results very useful for better understanding in this scientific area including implications for pollinator management. Their results can be considered as new insight into the aging of solitary bees.
I have several comments:
1) Literature citations are not numbered consecutively in the manuscript, which is inconsistent with directions for authors. Next formalities at line 69.
2) Picture 2F is not complete it is outside of the manuscript. I could see only basal oocyte volume graph.
3) You have noted that A. mellifera foragers older than 29 days experience a general decline in flight performance. But I did not find any information about longevity in Megachile rotundata females. Is the twenty-first day half of this bee life, two-thirds of it, or the last fifth of it? You mentioned only that this species constructs 8-12 brood cells in their lifetime. Is 21th day after emergence sufficient time frame for description of life of an old bee? Sorry, may be I overlooked this information/discussion about it but if it is missing it should be completed. I found only, contrary to your hypothesis, that M. rotundata did not experience apparent reproductive senescence within your experimental timeframe. Was this timeframe sufficient for description of biology of at least "an early old bee" in whom some senescence may have been potentially observed? That is crucial question because young bees do not necessarily show early signs of aging. My idea about of ​​this discussion is that it should be mentioned how long this bee species usually is living. To mention only number of built nests for whole life is vague if you did not counted sum of all built nests. I beleive that completing this information can improve interpretation of your results.
4) Contrary to previous note, your discussion and comparison your results with results by Richards is great. Your desing is more friendly to natural conditions for observation of the aging!
5) Lines 355-358: You found that 7 day old females had a shorter foraging window compared to day 14 or 21 females. You speculated that the shorter window of foraging in younger bees is due to physiological changes in flight muscles ... However, is it possible to speculate also about different temperature or weather conditions?
6) Methodology is largely well written, descriptively and exactly. Reference to source [95] (line 127) is possible to explain in details about used equations, however, L and r should be explained directly in the manuscript. Is not it so, please?
7) I am not sure if methodolody for Fig. 5A is sufficiently described. You wrote in 2.3 chapter: From the videos, we were able to determine the different phases of nesting (cell construction, provisioning, and cell completion) and calculate the number of brood cells constructed that day. There is no information about number of nests, therefore, I think that you have explained only Fig. 5B.
When these notes will be reflected I can recommend to publish this manuscript.
Author Response
Comment 1: Literature citations are not numbered consecutively in the manuscript, which is inconsistent with directions for authors. Next formalities at line 69.
Response 1: This issue has been appropriately addressed.
Comment 2: Picture 2F is not complete it is outside of the manuscript. I could see only basal oocyte volume graph.
Response 2: This will be addressed with the submission of high resolution.
Comment 3: 3) You have noted that A. mellifera foragers older than 29 days experience a general decline in flight performance. But I did not find any information about longevity in Megachile rotundata females. Is the twenty-first day half of this bee life, two-thirds of it, or the last fifth of it? You mentioned only that this species constructs 8-12 brood cells in their lifetime. Is 21th day after emergence sufficient time frame for description of life of an old bee? ... Was this timeframe sufficient for description of biology of at least "an early old bee" in whom some senescence may have been potentially observed?
Response 3: Literature survey indicates that generally the females live up to 21 days in the field. However, there is also some literature evidence that the female life span could be up to 28 days. In the laboratory conditions, we have also noted similarly (Pithan, et al., 2024. Journal of Insect Physiology, 104666.). We do believe that the 21 day timeframe is sufficient since a majority of the females do not survive beyond 21 days.
Comment 4: NA
Comment 5: Lines 355-358: You found that 7 day old females had a shorter foraging window compared to day 14 or 21 females. You speculated that the shorter window of foraging in younger bees is due to physiological changes in flight muscles ... However, is it possible to speculate also about different temperature or weather conditions?
Response: We have considered that possibility. However, when we took the temperature, humidity and other forecast data into account, we did not notice any significant departure from the day 7 during the other observed days.
Comment 6: Methodology is largely well written, descriptively and exactly. Reference to source [95] (line 127) is possible to explain in details about used equations, however, L and r should be explained directly in the manuscript. Is not it so, please?
Response 6: Thank you. We have added details to explain the equation better.
Comment 7: I am not sure if methodolody for Fig. 5A is sufficiently described. You wrote in 2.3 chapter: From the videos, we were able to determine the different phases of nesting (cell construction, provisioning, and cell completion) and calculate the number of brood cells constructed that day. There is no information about number of nests, therefore, I think that you have explained only Fig. 5B.
Response 7: Is the reviewer referring to Figure 3A-D? If that is the case, the figure shows the foraging activity of the females through days 7-21. 3D shows the number of brood cells that those females constructed during these days. If the reviewer is referring to figures 5A,B, then the figure 5A shows the number of nests that were collected from the nesting boxes over the number of observed females, and Fig. 5B is depicting the total number of brood cells from those nests over the number of observed females.
Reviewer 4 Report
Comments and Suggestions for Authors
- The research is well done, and the paper is very well written. I have only minor suggestions:
- Line 1 (the Title): I think that the title should better reflect the results of this research. For example: “No reproductive senescence in the pollinator, Megachile rotundata”.
- Line 15: In relation to the part “throughout the observed lifespan”. Since you observed only the 21-day period (not the entire lifespan), please consider replacing the word "lifespan" with "period of 21 days".
- Line 198. (The title of Figure 1.): In relation to the part “throughout their adult lifespan”, please consider replacing the words "their adult lifespan" with "the observed 21-day period".
- Line 204: (The title of Figure 1.): In relation to the part “with different letters were significantly different (P < 0.05)”, I must say I could not find “different letters”.
- Figure 2 (between lines 180 and 181) is only partially visible. In fact, the picture marked as (D) is hidden, but you have provided it separately, so I saw it. However, the graph marked as (F) is only partially visible, and I did not find it anywhere else.
- Lines 335-336: You should write that your results are in agreement with previous findings of no or little reproductive senescence (reviewed in 52; 64) that solitary insects often exhibit little to no reproductive senescence.
Author Response
Comment 1: Line 1 (the Title): I think that the title should better reflect the results of this research. For example: “No reproductive senescence in the pollinator, Megachile rotundata”.
Response 1: The authors have considered the reviewer's comment regarding the change of title. However, the authors feel that the change would be too strong because we believe that there is a possibility that senescence could occur in the individuals that survive beyond 21 days.
Comment 2: Line 15: In relation to the part “throughout the observed lifespan”. Since you observed only the 21-day period (not the entire lifespan), please consider replacing the word "lifespan" with "period of 21 days".
Response 2: Thank you for the comment. We have made appropriate changes to the manuscript.
Comment 3: Line 198. (The title of Figure 1.): In relation to the part “throughout their adult lifespan”, please consider replacing the words "their adult lifespan" with "the observed 21-day period".
Response 3: Thank you for the comment. We have made appropriate changes to the manuscript.
Comment 4: Line 204: (The title of Figure 1.): In relation to the part “with different letters were significantly different (P < 0.05)”, I must say I could not find “different letters”.
Response 4: This has been fixed in consultation with the journal in the newly uploaded high resolution figures.
Comment 5: Figure 2 (between lines 180 and 181) is only partially visible. In fact, the picture marked as (D) is hidden, but you have provided it separately, so I saw it. However, the graph marked as (F) is only partially visible, and I did not find it anywhere else.
Response 5: This has been fixed in consultation with the journal in the newly uploaded high resolution figures.
Comment 6: Lines 335-336: You should write that your results are in agreement with previous findings of no or little reproductive senescence (reviewed in 52; 64) that solitary insects often exhibit little to no reproductive senescence.
Response 6: Thank you. Due changes indicating that our results are in agreement with the cited reviews have been made in the manuscript.